# Differences of Training Structures on Stimulus Class Formation in Computational Agents

**Alexis Carrillo *** and **Moisés Betancort ***

Departamento de Psicología Clínica, Psicobiología y Metodología, Campus de Guajara, Universidad de La Laguna, Apartado 456, 38200 San Cristóbal de La Laguna, Spain
* Correspondence: alu0101534640@ull.edu.es (A.C.); moibemo@ull.edu.es (M.B.)

**Abstract:** Stimulus Equivalence (SE) is a behavioural phenomenon in which organisms respond functionally to stimuli without explicit training. SE provides a framework in the experimental analysis of behaviour to study language, symbolic behaviour, and cognition. It is also a frequently discussed matter in interdisciplinary research, linking behaviour analysis with linguistics and neuroscience. Previous research has attempted to replicate SE with computational agents, mostly based on Artificial Neural Network (ANN) models. The aim of this paper was to analyse the effect of three Training Structures (TSs) on stimulus class formation in a simulation with ANNs as computational agents performing a classification task, in a matching-to-sample procedure. Twelve simulations were carried out as a product of the implementation of four ANN architectures on the three TSs. SE was not achieved, but two agents showed an emergent response on half of the transitivity test pairs on linear sequence TSs and reflexivity on one member of the class. The results suggested that an ANN with a large enough number of units in a hidden layer can perform a limited number of emergent relations within specific experimental conditions: reflexivity on B and transitivity on AC, when pairs AB and BC are trained on a three-member stimulus class and tested in a classification task. Reinforcement learning is proposed as the framework for further simulations.

**Keywords:** stimulus equivalence; machine learning; matching to sample; artificial neural network

## 1. Introduction

Arbitrary dissimilar stimuli can be trained to be functionally related via conditional discrimination. Based on this idea, Sidman [1] carried out arbitrary Matching-To-Sample (MTS) experiments and found that functional properties can be transferred to other stimuli without explicit training; these emergent responses of reflexivity, symmetry, and transitivity are known as equivalence relations [2–4]. For example, during the training phase, a participant is rewarded when: an image of a cow is presented and the word "cow" is selected; an image of a cow is presented and the equivalent word in Spanish "vaca" is selected; an image of a cat is presented and the word "cat" is selected; an image of a cat is presented and its equivalent in Spanish "gato" is selected. After training, without reward, in the test phase, the word "cow" is presented, and the participant selects the word "vaca"; the word "gato" is presented, and the participant selects the image of a cat; the word "vaca" is presented, and the participant selects the word "vaca", as shown in Figure 1. In this example, six stimuli were trained to form two classes (1 and 2) with three members (A, B, and C) each, in an MTS procedure. Stimulus Class 1 consists of the image of a cat (A1), the word cat (B1), and the Spanish word "gato" (C1), while Stimulus Class 2 consists of the image of a cow (A2), the word "cow" (B2), and the Spanish word "vaca" (C2). The two classes were formed by training the A1–B1, A1–C1, A2–B2, and A2–C2 relations. The test phase showed emergent relations between the stimuli within the class members without explicit training: transitivity (B2–C2), symmetry (C1–A1), and reflexivity (C2–C2). In the example, the creation of a class corresponding to a picture and a written word relates

it to reading comprehension, meaning, and understanding. Furthermore, the relation established between the words of two languages illustrates the applicability of SE in foreign language acquisition. SE provides a framework in the experimental analysis of behaviour to study language, symbolic behaviour, and cognition, in a form of learning characterised as creative or generative. SE is also a common topic for interdisciplinary research linking behaviour analysis with linguistics and neuroscience [5–7].

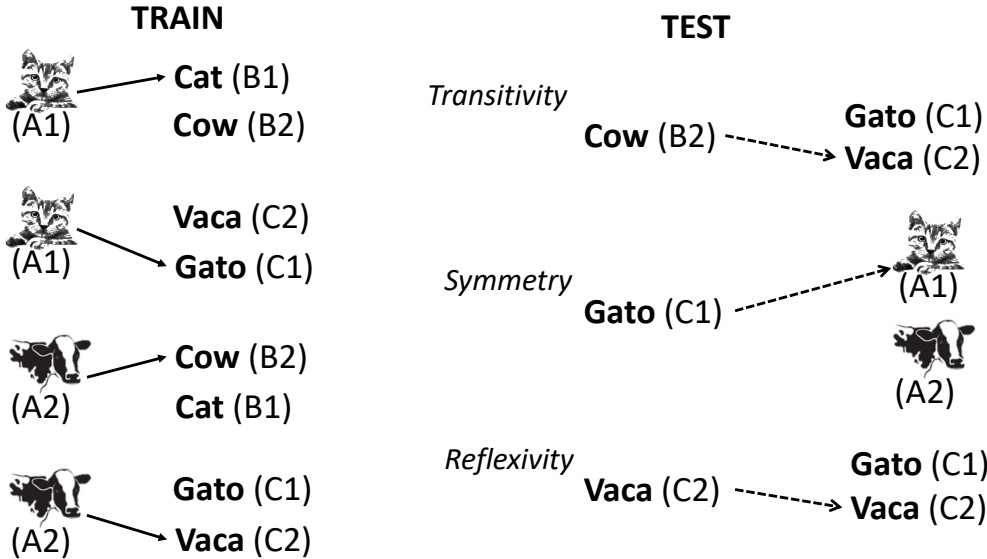

**Figure 1.** Example of 2 stimulus classes (1 and 2) with 3 members (A, B, and C). The left part displays the training phase, and the right part corresponds to transitivity symmetry and the reflexivity test.

A context of analytical units and stimulus control is required to understand the behavioural phenomenon of SE from the perspective of experimental behaviour analysis [8]. The behaviour of organisms can be analysed as environmental events linked by contingencies, if–then statements describing events that are true only under certain conditions [3]. Contingencies allow units of analysis to be formed according to the number of terms involved. The two-term unit, called operant reinforcement, is composed by the terms response–reinforcer: only a defined response (rsp1) produces a defined reinforcer (rf1) and not otherwise. In the three-term unit, called simple discrimination, a two-term unit is under the control of a discriminative stimulus: a response (rsp1) produces its reinforcer (rf1) only in the presence of a particular discriminative stimulus (B1). In the four-term unit, called conditional discrimination, a three-term unit is under the control of a conditional stimulus: a response (rsp1) produces its reinforcer (rf1) in the presence of a particular discriminative stimulus (B1) only if a conditional stimulus (A1) is present. The MTS procedure is also known as conditional discrimination training in which the conditional stimuli are called samples and the discriminative stimuli are called comparisons [1–4]. In the example in Figure 1, a trained conditional discrimination unit is: selecting option "X" produces its reinforcer (rf) in the presence of the word "cow" (B1) if the image of a cow (A1) is presented and is labelled as the A1–B1 relation. The A1–B1, A1–C1, A2–B2, and A2–C2 relations are conditional discrimination units explicitly formed in the training phase. However, the test phase showed that new conditional discrimination units were formed. One of these new untrained units is the B2–C2 relation: selecting option "X" produces its reinforcer (rf) in the presence of the word "vaca" (C2) if the word "cow" (B2) is presented. This means that a stimulus that was trained as a conditional stimulus can form new units that have not been previously trained, in which it can either control other three-term units or in which its role changes to a discriminative stimulus under the control of other stimuli. If a stimulus controls one member of the class, it affects all members of an equivalent stimulus

class [4]. Stimulus classes can be formed from any unrelated stimulus, arbitrarily assigned and functionally related regardless of its physical properties [2].

Sidman [2] affirmed that Stimulus Equivalence (SE) experiments provide an opportunity to study a type of stimulus generalisation in which participants functionally match dissimilar stimuli that have never been directly related. Understanding and correctly reproducing the conditions under which these emergent relations manifest themselves are fundamental knowledge for empirical, theoretical, and applied inquiry. Arntzen [5] presented a set of procedural variables of relevance for methodological considerations in SE research. The Training Structure (TS) [9–12] describes the method of selecting which stimulus pairs will be related by training in the MTS procedure [2,10,11] and which pairs will be in the test for emergent relations. TSs can be Linear Series (LSs), Many-To-One (MTO), and One-To-Many (OTM). In establishing an ($C = 1$) equivalent class with four ($M = 4$) members: A, B, C, and D, it is necessary to train $C(M - 1) = 1(4 - 1) = 3$ pairs and evaluate $C(M - 1)^2 = 1(4 - 1)^2 = 9$ emergent relations of transitivity and symmetry. In MTO, sample stimuli (B, C, and D) are trained with one comparison stimulus (A), so the training pairs are BA, CA, and DA. In OTM, one sample stimulus (A) is trained with many comparisons (B, C, and D), corresponding to training pairs AB, AC, and AD. In an LS training structure, a sample stimulus is shifted as a comparison to a sample of another pair. The pairs in the LS are AB, BC, and CD. In the AB pair, A is the sample and B is the comparison; then, B is the sample and C is the comparison in the BC pair, and C is the sample and D is the comparison for the CD pair [5,9,11]. Figure 2 shows the trained relations: AB, BC, and CD for the LS; AB, AC, and AD for OTM; BA, CA, and DA for MTO, and the emergent relations evaluated: reflexivity, transitivity, and symmetry.

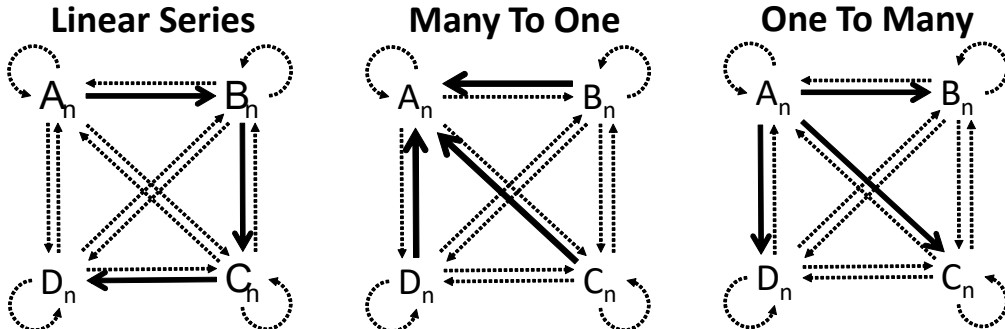

**Figure 2.** Members (A, B, C, and D) of the class (*n*) are trained according to the training structure, as shown by the solid arrows. The emergent relations of reflexivity, transitivity, and symmetry for the evaluation are shown as dashed arrows.

Fifty years of research on stimulus equivalence have covered a wide range of topics from basic to applied, both in humans and non-human animals [4,6], and also in computational simulations often involving an Artificial Neural Network (ANN) to replicate SE in computational agents [7]. The ANN is a bio-inspired algorithm based on a simplification of the functioning of neurons and their relationship to information flow and processing. A single artificial neuron is called a perceptron, and interconnected, layered artificial neurons that learn simultaneously through the backpropagation of errors [13] can perform a wide variety of complex tasks [14–17]. Deep Learning (DL) is the field of Machine Learning (ML) related to the research and development of those complex ANNs [14]. As a part of computer science, ML seeks to develop algorithms that can improve their performance on a specific task, without the need for programming their execution, by mapping functional relations between variables in datasets. ML algorithms are classified according to the type of feedback signal available into three categories: supervised learning, unsupervised learning, or Reinforcement Learning (RL) [15,18,19]. In supervised learning, there is a dataset for which the expected outcome is known; the algorithm is expected to infer a function from the labelled training data that maps an input to an output; if the target

variable is continuous, the algorithms perform a regression task [20], and if it is categorical, it is called a classification task [21]. Unsupervised learning algorithms are not given an expected outcome value, so they identify underlying patterns in the data; tasks may consist of identifying relations between elements in the set, as in clustering tasks, where categories are assigned based on the properties of the elements or dimensional reduction may also be performed to find a way to represent the data more compactly [15,16]. In RL, an agent embedded in an environment performs an action in a given time step ($A_t$) that changes the state of the environment ($S_t$). This new state ($S_{t+1}$) and a reward signal ($R_{t+1}$) are returned as the input to the agent, which performs a new action. The agent must learn how to map situations to actions in order to maximise the reward signal [14,17,22].

Computational models of psychological phenomena contribute to the advancement of both theory and research in psychology. The simulation of SE is an opportunity to explore experimental conditions that are difficult to achieve in real conditions with participants due to limitations such as costs, time, experimental control, ethical implications, or practicality. Experimenting and testing various configurations of variables in a short period of time can provide significant contributions, with particular emphasis on parameters for establishing stimulus equivalence [5,23,24]. The aim of this paper was to analyse the effect of different TSs on class formation in a simulation with ANNs as the computational agents performing a classification task.

*Related Work*

RELNET [23], a feedforward fully connected ANN with eight units in the hidden layer, is known as the first reference to the use of an ANN in the simulation of emergent relations in the context of relational frame theory. Criticisms of RELNET [7,25,26] point to the mixing of information from training and test data via the sample-marking duplicator, a part of the input encoding. Tovar and Torres-Chavez [25] also implemented a feedforward network with 6 hidden units simulating compound stimuli procedures with 2 units in the output layer, as yes/no responses. A replication of this model was carried out by Vernucio and Debert [27] with the modification of a single unit in the output layer, simulating go/no-go responses during compound stimulus procedures. Ninness et al. [26] proposed another feedforward network called emergent virtual analytics (EVA), without sample-marking duplicators and independent of extraexperimental training values. A broader view of computational models of stimulus equivalence was presented by Tovar et al. [7], classifying the models into four groups: feedforward networks, self-organising maps [28], biologically inspired neural networks [29], and reinforcement learning [24,30]. The feedforward network RELNET [23] and those of Tovar and Torres-Chavez [25], Vernucio and Debert [27], and EVA [26] have in common three limitations: the use of a bit to represent a stimulus in the input encoding, training only the LS structure, and having a reduced number of units in the hidden layer. In this regard, this paper aimed to contribute by proposing a connectionist model without a sample-marking duplicator, improvements in the encoding of the stimulus as an input vector for the ANN, a comparative analysis of multiple TSs, and testing of four ANN architectures with variations in both the number of units and the number of hidden layers.

## 2. Materials and Methods

### 2.1. MTS Procedure

The experimental conditions followed the specifications of Arntzen [5,9]. Each one of the three TSs: LS, OTM, and MTO, were established as experimental conditions with simultaneous matching to the presentation of the sample and comparisons. The pairs' distribution by the TSs was as follows: in the LS: training (AB, BC), reflexivity (AA, BB, CC), symmetry (BA, CB), and transitivity (AC, CA); MTO with C as the node: training (AC, BC), reflexivity (AA, BB, CC), symmetry (CA, CB), and transitivity (AB, BA); OTM with A as the node: training (AB, AC), reflexivity (AA, BB, CC), symmetry (BA, CA), and transitivity (CB, BC); explicit training pairs: TXTY, TXTZ, TXTX, TYTY, TZTZ, TYTX, TYTZ, TZTX, TZTY.

For each TS, a total of 45 experimental pairs were created and formed the following groups: Training pairs included both the TS and explicit training. Test pairs were divided by the emergent relation type: reflexivity, symmetry, and transitivity. Experimental trials were selected from the full combination of the 15 stimuli in 4 positions (sample, Comparisons 1, 2, and 3) $15^4 = 50,625$. First, an individual trial was created, and then, each pair of sample comparisons was evaluated. A combination was considered a valid trial if there was only one experimental pair between the 3 comparison stimuli and the sample. If there was no pair in any of the comparisons, it was considered a no-response trial. As a result, valid trials were assigned to the following groups: 3456 to training, 3888 to explicit training, 5184 to reflexivity, 3456 to symmetry, and 3456 to transitivity, 25,920 were separated as no-response trials from the rest.

A dataset was proposed for an MTS task with 1 sample and 3 comparison stimuli. The stimulus set consisted of four 3 member classes; the members' labels were A, B, and C, and the classes corresponded to the numbers 1 to 4 (A1, B1, C1; A2, B2, C2; A3, B3, C3; A4, B4, C4). An additional fifth class was created for training of all relations (TX, TY, TZ), taking into account Barnes's proposal [23] that explicit training increases agent performance, which was used in the simulations of Tovar [25] and Vernucio and Derbert [27]. Each experimental trial consisted of the presentation of the sample and 3 comparison stimuli. The agent must indicate either the position of the response to the sample or none when there is no answer requested. No response trials were included to balance the dataset.

### 2.2. Stimulus Encoding

As the input data for the agents, each stimulus was encoded as a sequence of 15 digits filled with zeros, except for a value of one at a given position for unique identification. The stimulus label and the corresponding coding are shown in Table 1. The entry for an experimental trial consisted of the concatenation of the sample, Comparison1, 2, and 3 digits of the trial, in a 60-digit sequence. For example, a training trial B1–C1 consisted of the sample B1; Comparison 1 is B2; Comparison 2 is C1; Comparison 3 is C3. B1 is coded 000001000000000, B2 000000100000000, C1 000000000010000, and C3 000000000000100. The input data for this specific trial were: 000001000000000000000100000000000000000010000000000000000100. The response encoding consisted of a 3-digit sequence where one marks the position of the response, and the omission of the response is encoded as a sequence of 3 zeros, as presented in Table 2. In the above example, the response is the second comparison stimulus C1, which is encoded as 010.

**Table 1.** Encoding of the stimulus for input values.

| Stimulus | Digit Sequence |
|---|---|
| A1 | 100000000000000 |
| A2 | 010000000000000 |
| A3 | 001000000000000 |
| A4 | 000100000000000 |
| TX | 000010000000000 |
| B1 | 000001000000000 |
| B2 | 000000100000000 |
| B3 | 000000010000000 |
| B4 | 000000001000000 |
| TY | 000000000100000 |
| C1 | 000000000010000 |
| C2 | 000000000001000 |
| C3 | 000000000000100 |
| C4 | 000000000000010 |
| TZ | 000000000000001 |

**Table 2.** Encoding of the response for output values.

| Response | Digit Sequence |
|---|---:|
| 1 | 100 |
| 2 | 010 |
| 3 | 001 |
| none | 000 |

### 2.3. ANN Agents

All ANNs used as agents had the same 60 input and 3 output units; their difference lied in the variation in the number of hidden layers and the number of units per layer. Defining the architecture of an ANN means that the researcher establishes the number of units and how those units are connected in the overall structure, which has implications for the ability to learn a specific function or generalise it [14]. Four agents with different architectures were established: (a) a single hidden layer that was 20000 units wide (Wide1); (b) two hidden layers of 2000 units each (Wide2); (c) 6 hidden layers of 105, 90, 75, 60, 45, and 30 units (Deep1); (d) 10 hidden layers of 100 units each (Deep2). RELU was the activation function of the units in the hidden layers [31], and softmax was the activation function in the output layer. Adam was the optimisation method [32,33]. A graphical representation of the architectures is presented in Figure 3.

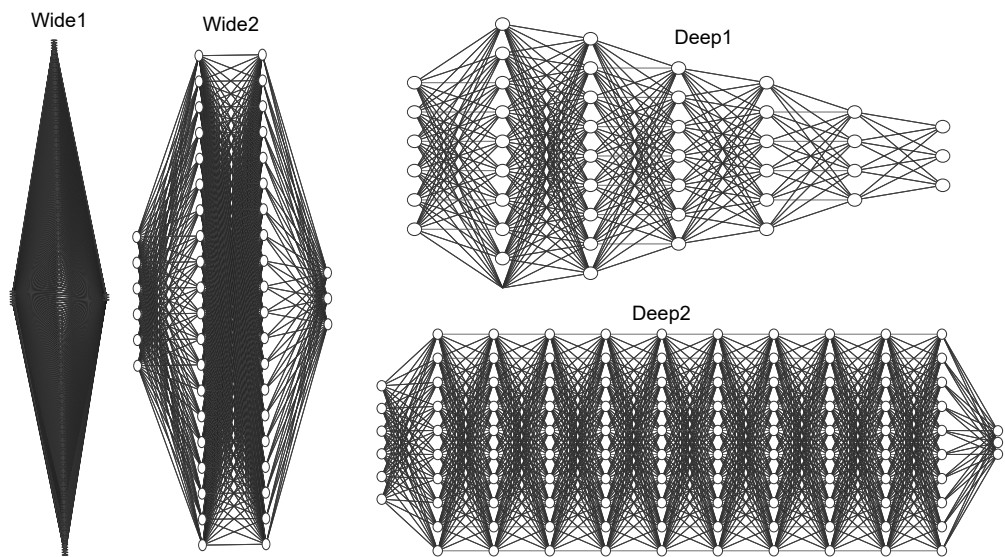

**Figure 3.** Network architectures for the 4 agents. All ANNs have the same 60 input and 3 output units. The number of units for the hidden layer is: Wide1 (20,000); Wide2 (2000, 2000); Deep1 (105, 90, 75, 60, 45, 30); Deep2 (100, 100, 100, 100, 100, 100, 100, 100, 100, 100).

### 2.4. Simulations

Twelve simulations were run from the combination of the 4 agents and 3 training structures. From the ML perspective, this is a multi-label classification task [15,17,21]. The coded trials were used as the input data and the coded responses as the target values. To balance the dataset, the same number of trials was randomly selected from the non-response trials. Training pairs, explicit training pairs, and non-response pairs formed the training set for a total of 9272 randomly ordered trials. The F1-weighted average score was the selected performance metric [34], ranging from 0 as the worst possible score to 1 as the best possible score, implying a completely correct response and the correct omission of non-response trials. The simulations were run on a desktop PC with the Windows 10 operating system, 24 Gb RAM, 1TB SSD storage, and a 3.70 GHz AMD Ryzen 7 2700X eight-core processor. To develop the code for the simulations, we used the Python V.3.7 programming language [35], Anaconda Distribution V.4.12 [36], and the scikit-learn library [37,38].

### 3. Results

The F1-score averages of the stimulus pairs, grouped by evaluation sets, are presented in Table 3 for each agent. The four ANN architectures were used as a different agent for each of the three TSs. The Wide1, Wide2, and Deep1 architectures scored highest in the training phase in all TSs. The reflexivity pairs in the LS reached around 50%, while it was above 90% in transitivity for Wide1, Wide2, and Deep1 architectures. The OTM and MTO test pairs performed the lowest, in contrast to the LS.

**Table 3.** Agents' F1-score by the TS and pairs' group.

| TS | Pairs' Group | Wide1 | Wide2 | Deep1 | Deep2 |
|----|--------------|-------|-------|-------|-------|
| LS | Train | 1.00 | 1.00 | 1.00 | 0.00 |
| | Reflexivity | 0.50 | 0.50 | 0.43 | 0.00 |
| | Symmetry | 0.00 | 0.00 | 0.00 | 0.00 |
| | Transitivity | 1.00 | 1.00 | 0.90 | 0.00 |
| | Transitivity–Symmetry | 0.00 | 0.00 | 0.00 | 0.00 |
| OTM | Train | 1.00 | 1.00 | 1.00 | 0.00 |
| | Reflexivity | 0.00 | 0.00 | 0.00 | 0.00 |
| | Symmetry | 0.00 | 0.00 | 0.00 | 0.00 |
| | Transitivity | 0.00 | 0.00 | 0.00 | 0.00 |
| MTO | Train | 1.00 | 1.00 | 1.00 | 0.33 |
| | Reflexivity | 0.00 | 0.00 | 0.00 | 0.10 |
| | Symmetry | 0.00 | 0.00 | 0.00 | 0.00 |
| | Transitivity | 0.00 | 0.00 | 0.00 | 0.00 |

There was a performance difference for both reflexivity and transitivity in the LS, which was superior to MTO and OTM. An exploration of the evaluation for stimulus-separated reflexivity pairs is presented in Table 4 and shows that the performance was different for Stimulus Member B in all four classes of the reflexivity evaluation. In the Wide1 and Wide2 architectures, the performance was 100%; in Deep1, it was lower; in Deep2, it was 0%. For the transitivity pairs, Table 5 shows the performance separated by pairs. Wide1 and Wide2 achieved transitivity for AC pairs, but not for CA pairs. Deep1 achieved partial performance on transitivity for different classes, and Deep2 did not perform on transitivity.

**Table 4.** Agents' F1-score by pair in the LS reflexivity test.

| | Architecture | | | |
|------|-------|-------|-------|-------|
| Pair | Wide1 | Wide2 | Deep1 | Deep2 |
| A1A1 | 0.00 | 0.00 | 0.00 | 0.00 |
| A2A2 | 0.00 | 0.00 | 0.00 | 0.00 |
| A3A3 | 0.00 | 0.00 | 0.00 | 0.00 |
| A4A4 | 0.00 | 0.00 | 0.00 | 0.00 |
| B1B1 | 1.00 | 1.00 | 1.00 | 0.00 |
| B2B2 | 1.00 | 1.00 | 0.67 | 0.00 |
| B3B3 | 1.00 | 1.00 | 1.00 | 0.00 |
| B4B4 | 1.00 | 1.00 | 0.67 | 0.00 |
| C1C1 | 0.00 | 0.00 | 0.00 | 0.00 |
| C2C2 | 0.00 | 0.00 | 0.00 | 0.00 |
| C3C3 | 0.00 | 0.00 | 0.00 | 0.00 |
| C4C4 | 0.00 | 0.00 | 0.00 | 0.00 |

**Table 5.** Agents' F1-score by pair in the LS transitivity test.

| Pair | Architecture | | | |
|------|------|------|------|------|
| | **Wide1** | **Wide2** | **Deep1** | **Deep2** |
| A1C1 | 1.00 | 1.00 | 1.00 | 0.00 |
| A2C2 | 1.00 | 1.00 | 0.67 | 0.00 |
| A3C3 | 1.00 | 1.00 | 1.00 | 0.00 |
| A4C4 | 1.00 | 1.00 | 0.67 | 0.00 |
| C1A1 | 0.00 | 0.00 | 0.00 | 0.00 |
| C2A2 | 0.00 | 0.00 | 0.00 | 0.00 |
| C3A3 | 0.00 | 0.00 | 0.00 | 0.00 |
| C4A4 | 0.00 | 0.00 | 0.00 | 0.00 |

The data suggested differences according to the type of TS. In MTO and OTM, there was no evidence of the emergent relations of reflexivity or symmetry, in contrast to the LS in the Wide1 and Wide2 agents. As described by Arntzen [5], the B member stimulus in LS training trials acts as a comparison in the pair AB and as a sample in the pair BC. This change of position during training could be mapped as an abstract feature by the ANN, which allows it to establish the emergent relation of reflexivity (B–B) and transitivity (A–C), as shown in Figure 4. It is said that an equivalence class has been established when the emergent relations of reflexivity, symmetry, and transitivity are manifested [4], independent of the TS. However, the evaluation of the whole set of emergent relations in our ANN agents showed that these requirements were not fulfilled, despite partially manifesting the emergent relations of reflexivity and transitivity.

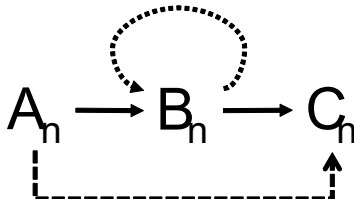

**Figure 4.** The emergent relations of transitivity (AC) and reflexivity (BB) observed in the Wide1 and Wide2 agents are shown in dashed arrows. The LS trained relations (AB, BC) are shown in solid arrows.

## 4. Discussion

The analysis of the implications of the findings was divided by field of knowledge: first, from the perspective of the experimental analysis of behaviour, in terms of training structure and related work; second, from the ML perspective, the analysis relates to ANN architectures, classification tasks, feature space representations, and contributions to the ML field.

### 4.1. Behavioural Analytical Perspective

The experiments reviewed indicated that the establishment of equivalent classes is independent of the TS, although LS training tends to be less effective than MTO and OTM [12,39]. Our simulations differed from this pattern. None of the agents demonstrated stimulus equivalence in either MTO or OTM. The Wide1 and Wide2 agents in the LS responded according to the transitivity emergent relation in the pair AC and the reflexivity emergent relation in the pair BB. In the LS, the B stimulus served both as the sample and comparison and may possibly be related to the emergent relations manifested. Future research may contribute to elaborating on these findings, as Arntzen [5] also suggested. Some of the main innovative aspects of this research were the use of three TSs compared to the simulations of Tovar [25], Vernucio and Derbert [27], and EVA [26], trained only with the LS and the use of a concatenation of encoded stimuli, instead of a sequence of digits

representing one stimulus per bit. Furthermore, the suggested encoding allowed the control of the stimulus position and the presentation of more than two stimuli in a simultaneous MTS procedure. This control allowed us to evaluate all emergent relations. The limitations of the input encoding in the Tovar [25], Vernucio and Derbert [27], and EVA [26] models restricted them to assessing only the AC emergent relation, which they reported as a success in transitivity testing. The results of this research were consistent with those findings; however, our extended tests gave us a broader view, and we concluded that only transitivity in the AC pair and reflexivity in the BB pair, trained on an LS, are not evidence of equivalence class formation in our agents. We propound that the presented encoding and the extended testing can be implemented for a better understanding of the performance of other models that differ from feedforward ANN architectures, such as EPS [24,30], or that of Tovar and Westerman [29], or SOM [28]. One possible reason for the observed emergent relations could be that explicit training pairs incorporate extraexperimental training values in the agents [7,23,25–27] as a guide in the formation of equivalence classes. If explicit training favours the formation of stimulus classes, it is expected that none of the emergent relations observed in the agents can manifest themselves in the absence of such training. The 12 simulations were run again without explicit training and excluding trials with TX, TY, and TZ, both in training and testing. The performance of the agents did not change: Wide1 and Wide2 manifested the same emergent relations. This is an important difference in our model: non-extraexperimental training values are required to manifest emergent relations.

Previous studies involving connectionist models failed to replicate human performance, possibly due to technology limitations associated with computational power, algorithms, ML tasks and types of learning that were not available at the time the first simulations were published. DL and RL have been extensively investigated especially during the last decade [14,15,17,22] and are the most-viable alternatives to achieve SE in computational agents. New approaches with an updated ML perspective have the potential of yielding promising results. Mofrad et al. [24,30] demonstrated that RL offers a more suitable framework for simulation and experimentation of behaviour analysis because it is directly inspired by behavioural theory and concepts. Further research within the RL framework is needed.

### 4.2. ML Perspective

The BB and AC emergent relations were observed in Wide1 and Wide2, but not in Deep1 or Deep2 agents. Based on the universal approximation theorem, a feedforward network with a sufficiently large single hidden layer can represent any function [14], but the number of units may be so large that there is not enough computational power to train it. Theoretically, it is possible for an ANN to establish equivalence relations, but in practice, the learning conditions and elements of experimentation can affect performance. This idea was applied in the Wide1 and Wide2 architectures. On the other hand, the Deep1 and Deep2 architectures were defined based on the concept that ANNs with a large number of hidden layers could represent more abstract features. In our simulations, agents with the Wide1 and Wide2 architectures performed better than the Deep1 and Deep2 agents, but only in the LS. This suggests that a large indeterminate number of units and few hidden layers are the most-likely scenario for an ANN architecture that evidences stimulus equivalence. However, there is no consensus on how many layers a neural network needs to be considered sufficiently deep, and therefore, 10 hidden layers may not be deep enough to exhibit these abstraction properties. Analysing the impact of different architectures or finding an optimal architecture that uses as few units as possible comprises new research ideas and simulation experiments.

An MTS procedure was adopted in the ANN as a multiclass classification task. The challenge for agents is to map conditional discriminative units as functions, using only a coded sequence of a set of stimuli as the input and a coded response as the output. In training, the agent is expected to identify the stimuli, label them as a conditional stimulus or discriminative stimulus, locate the discriminative stimulus among the com-

parisons, and return the correct response. In this study, all four agents in the three TSs scored highest in the training phase. The agents showed potential to produce abstract representations of conditional discrimination units. The arbitrary MTS procedure implies that the criterion for forming a set of stimuli is independent of their physical properties and, therefore, will not share any similarity. The performance of the agents in the training phase suggests that ANNs are capable of extracting features and abstractly representing conditional discrimination units in feature spaces that can be explored with the method proposed by Kansizoglou et al. [31]. In contrast, the failure to pass most of the evaluations in the test phase indicates that our agents are not fully capable of extending the properties of stimulus control to all members of the equivalence class. However, passing the test for reflexivity in the BB pair and transitivity in the AC pair in the LS showed that ANNs have the potential to form stimulus equivalence classes. It is not clear what the conditions or what the requirements are for an ANN to demonstrate the formation of equivalent classes. Possible candidates for agents in new simulations may be generative networks [14].

Can other ML algorithms perform conditional discrimination and manifest emergent relations? To answer this question, simulations were performed with the LS as the TS with seven machine learning algorithms: k-nearest neighbours, support vector machine with linear function kernel, support vector machine with the radial basis function kernel, random forest, adaptive boosting, naive Bayes, and quadratic discriminant analysis, from the scikit-learn library [38]. For the evaluation of the training pairs, k-nearest neighbours, support vector machine with the radial basis function kernel, and quadratic discriminant analysis obtained results above 0.97 on the F1-score, but only k-nearest neighbours obtained an F1-score above 0 in transitivity (0.457) and reflexivity (0.198). In the evaluation of the scores of the emergent relation pairs, it was observed that, in the reflexivity pair BB and in the transitivity pair AC, the F1-scores were 0.457 for both. Support vector machine with the radial basis function kernel and quadratic discriminant analysis share with ANNs that they perform feature space transformations, but this is not the case in k-nearest neighbours. A comparative analysis is beyond the scope of this paper. It is suggested that future research should further study the differences in the performance of different ML algorithms in manifesting emergent relations.

The theoretical results in ML can be applied to the study of learning as a psychological phenomenon [19], which opens a line of applied research. Alternatives for future research are to use a different approach to the ML task and to implement a different architecture than the fully connected feedforward ANN type [7]. It is not only necessary to choose the right ML task, but also to choose the right architecture. The classification task is part of supervised learning; alternatives to this approach are unsupervised learning, as used in clustering algorithms, semi-supervised learning, or RL. The most-promising approach for this is the framework of RL, which it is inspired by Thorndike's effect law [14–17,22]. A successful RL-based approach is the projective simulation framework, extended and improved by Mofrad et al. [24,30]. Alternatives to feedforward fully connected ANNs are recurrent networks, autoencoders, and generative networks [14].

An abstract representation of conditional discrimination units can be considered a significant contribution to the ML field. To state that an ANN can have abstract representations of conditional discrimination units implies that it is assembling and storing in its feature space a series of if–then statements that relate four elements: a reward, an action, a discriminative stimulus, and a conditional stimulus. This has some similarities to a symbolic AI system [40], but the rules are automatically assembled by an ANN. Future research could explore whether ANNs are actually constructing such abstract representations of rules and logical sentences from the conditional discrimination units.

Simulating SE in ANNs is part of a growing line of research analysing psychological phenomena in DL models, conceived of as the scientific study of behaviour exhibited by intelligent machines in the interdisciplinary field of machine behaviour [41]. General intelligence and abstract reasoning are considered important for understanding the ways in which ANNs are able to process abstract concepts [42]. Abstraction and reasoning

corpora [43] have been proposed to measure in ML models general fluid intelligence. Ritter et al. carried out cognitive tests on one-shot learning models trained on ImageNet [44], proposing the study of artificial cognitive psychology. Cognitive tasks such as decision-making, information search, deliberation, and causal reasoning abilities have been tested on GPT-3 [45].

## 5. Conclusions

The present study attempted to simulate SE in neurocomputational agents following the approaches of Barnes [23], Ninnes [26], Tovar and Torres-Chavez [25], and Vernucio and Derbert [27], but with differences in the input encoding, the proposal of 4 architectures instead of 1, and 3 TSs vs. the use of 1 in the aforementioned publications. Although the agents did not demonstrate equivalence class formation, emergent relations were found in the BB and AC pairs when trained with the LS. This study showed that methodological considerations such as TSs can affect the performance of an ANN. The use of more possibilities in the number of agents, their complexity in architecture, and variations in experimental conditions suggest that larger simulations can provide a broad perspective that could help clarify unanswered questions related to the experimental conditions affecting the establishment of equivalence classes.

Our proposed coding of the input values allowed a more thorough evaluation of the equivalence relations than previous simulations and may also contribute to a better understanding of the phenomenon in future research. Now, there are still some questions that remain open as a result of this research: Can ANNs create abstract representations of structured logic sentences? Why can ANNs only establish a limited number of emergent relations in the LS? What makes Stimulus B in the LS manifest emergent relation reflexivity? Can other ANN architectures manifest equivalence class formation? What results can be obtained from implementing these simulations as a reinforcement learning task?

**Author Contributions:** Supervision, M.B.; writing, A.C. All authors have read and agreed to the published version of the manuscript.

**Funding:** This research received no external funding.

**Data Availability Statement:** The simulations' code can be found in https://github.com/Yagwar/stim_eq/tree/master/MDPI_TS_SE_Article, accessed on 9 March 2023.

**Conflicts of Interest:** The authors declare no conflict of interest.

## Abbreviations

The following abbreviations are used in this manuscript:

| | |
|---|---|
| SE | Stimulus Equivalence |
| ANN | Artificial Neural Network |
| ML | Machine Learning |
| DL | Deep Learning |
| RL | Reinforcement learning |
| MTS | Matching To Sample |
| TS | Training Structure |
| MTO | Many-To-One |
| OTM | One-To-Many |
| LS | Linear Structure |

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
