# Peer review of "Differences of Training Structures on Stimulus Class Formation in Computational Agents"

_mti, doi:10.3390/mti7040039_

Round 1

Reviewer 1 Report

Overview

The paper at hand proceeds with an experimental study based on 12 simulations in order to study stimulus equivalence of three-member classes in Artificial Neural Networks (ANNs). My general concerns, described in detail through the comments below, are summarized as follows. Moderate English modifications and method clarifications are required. Although the paper puts ANNs at the center of their study, recent works and findings in the field of deep learning are completely missing. The authors claim in their discussion: “Findings suggests that, in general ANN are not capable of generalising enough to equivalence response.” However, no comparative study on the same or similar experiment or related recent works is provided, in order for the reader to safely conclude to such a conclusion. I believe the paper could be highly benefited from a comparative study with other machine learning and/or conventional methods. Comparison results of ANNs on similar simulations provided in the recent bibliography could also be included. Overall, the paper is well organized and presents a quite interesting topic with promising results for further investigation.

Comments

Line 1: "organisms responds" => "organisms respond"

Line 2: " has trying to" => "has attempted to"

Line 3: " Results has been promissory" => " Results have been promissory"

Line 4: Abstract should not contain any references. Please remove Ref. [1] from the abstract.

Line 5: "training structures (TS)" => "Training Structures (TS)" to keep concise with the rest acronyms of the paper.

Line 12: "on AC when pairs" => "on AC, when pairs"

Line 12: "trained of a three-member class" => "trained on a three-member class"? it is not clear what the authors imply at this phrase.

Line 13: "Reinforcement Learning" => "Reinforcement learning"

Line 20: "that SE" => "that Stimulus Equivalence (SE)" first appearance within the body of the manuscript.

Line 22-23: "Understand and correctly replicate..." => "Understanding and correctly replicating..."

Line 23: "emergent relations manifests" => "emergent relations manifest"

Line 26: " the training structure (TS)" => " the Training Structure (TS)"

Line 27: Please provide explanation or reference for MTS procedure

Line 49: “those complex ANN” => “those complex ANNs”

Line 53-54: Ref needed: Jordan, Michael I., and Tom M. Mitchell. "Machine learning: Trends, perspectives, and prospects." Science 349.6245 (2015): 255-260.

Line 56-57: “ if the target variable is continuous, the algorithms perform a regression task, and if is categorical it is called a classification task.” recent studies on both regression and classification models are missing. Please add references:

Lai, Yuandu, et al. "Exploring uncertainty in regression neural networks for construction of prediction intervals." Neurocomputing 481 (2022): 249-257. Chauhan, Vikas, and Aruna Tiwari. "Randomized neural networks for multilabel classification." Applied Soft Computing 115 (2022): 108184.

Line 120: "A graphic" => "A graphical"

Line 163: "on angent's" => "on agent's"

Line 177: "an perform better" => "and perform better"

Line 181-182: “a Neural Net in order to be considered as deep, and therefore” => “a neural network in order to be considered sufficiently deep and therefore”

Line 195: “Findings suggests that, in general ANN are” => “Findings suggest that in general, ANNs are”

Line 195-196: This is quite a strong statement, yet no comparative study or comparative related works are presented in order to justify such a claim. In general, the authors should compare the performance of the exploited ANNs with other methods either machine learning-based or conventional ones in order to complete their study. Possibly, the idea of unsupervised or reinforcement learning could be studied, as well.

Line 199-201: “Given the nature of the classification task, … ANN agents to generalize.” The above describes the ANN’s feature discrimination capacity which is a recently proven statement explaining how ANNs discriminate their deep feature space based on the target classes and the learned weights. Hence, a geometrical interpretation of the classes loci is also realized all described in Kansizoglou, et al. "Deep feature space: A geometrical perspective." IEEE Transactions on Pattern Analysis and Machine Intelligence 44.10 (2021): 6823-6838. The authors could refer to this analysis and exploit it as future work to better analyze and further enhance the generalization capacity of their ANNs in the classification task, along with unsupervised and/or reinforcement learning. Based on that, deep learning benchmark approaches have been introduced for improved generalization. Please refer to ArcFace, HASeparator, and Sphereface and discuss about them in ANNs' discrimination and generalization capacities.

Finally, re-read the entire manuscript and proceed with further English modifications.

Author Response

Although the paper puts ANNs at the center of their study, recent works and findings in the field of deep learning are completely missing.

Response: A review of related and recent works on interdisciplinary field of machine behaviour was added in the discussion section.

The authors claim in their discussion: “Findings suggests that, in general ANN are not capable of generalising enough to equivalence response.” However, no comparative study on the same or similar experiment or related recent works is provided, in order for the reader to safely conclude to such a conclusion.

I believe the paper could be highly benefited from a comparative study with other machine learning and/or conventional methods.

Comparison results of ANNs on similar simulations provided in the recent bibliography could also be included.

Response: A related work subsection was added to the introduction and comparative analysis of was added in discussions.

Overall, the paper is well organized and presents a quite interesting topic with promising results for further investigation.

Response: A more detailed analysis of was added to provide more context related with the importance of this paper.

Line 53-54: Ref needed: Jordan, Michael I., and Tom M. Mitchell. "Machine learning: Trends, perspectives, and prospects." Science 349.6245 (2015): 255-260.

Response: Reference added 

Line 56-57: “ if the target variable is continuous, the algorithms perform a regression task, and if is categorical it is called a classification task.” recent studies on both regression and classification models are missing. Please add references:

Lai, Yuandu, et al. "Exploring uncertainty in regression neural networks for construction of prediction intervals." Neurocomputing 481 (2022): 249-257. Chauhan, Vikas, and Aruna Tiwari. "Randomized neural networks for multilabel classification." Applied Soft Computing 115 (2022): 108184.

Response: References added 

Line 199-201: “Given the nature of the classification task, … ANN agents to generalize.” The above describes the ANN’s feature discrimination capacity which is a recently proven statement explaining how ANNs discriminate their deep feature space based on the target classes and the learned weights. Hence, a geometrical interpretation of the classes loci is also realized all described in Kansizoglou, et al. "Deep feature space: A geometrical perspective." IEEE Transactions on Pattern Analysis and Machine Intelligence 44.10 (2021): 6823-6838. The authors could refer to this analysis and exploit it as future work to better analyze and further enhance the generalization capacity of their ANNs in the classification task, along with unsupervised and/or reinforcement learning. Based on that, deep learning benchmark approaches have been introduced for improved generalization. Please refer to ArcFace, HASeparator, and Sphereface and discuss about them in ANNs' discrimination and generalization capacities.

Response: A more detailed explanation of the Stimulus equivalence phenomenon was added in the introduction and specific analysis concerning to emergent relations and feature space representations of conditional discrimination units were added, considering the reference suggested. 

Moderate English modifications are required .

Response: After modifications, a revision of english was carried out.

Reviewer 2 Report

Authors proposed the Differences of Training Structures on the Stimulus Class Formation in Computational Agents. After reading the whole paper, following observation have been made.

1.The structure and layout of the paper should be consistent throughout.

2. The usage of the English language should be improve in the paper.

3. The novelty of the paper is limited and should be improved.

4. The title of the figures should be improved.

5. A comparative analysis with the existing work should be incorporated in the paper.

6. Reference section should be strengthened with recent ones. 

Author Response

1.The structure and layout of the paper should be consistent throughout.

Response: Modifications to the structure and subsections were added to improve the consistency of the paper.

2. The usage of the English language should be improve in the paper.

Response: After modifications, a revision of english was carried out.

3. The novelty of the paper is limited and should be improved.

Response: A more detailed analysis of was added to provide more context related with the importance of this paper.

4. The title of the figures should be improved.

Response: Modifications to the titles and descriptions were carried out.

5. A comparative analysis with the existing work should be incorporated in the paper.

Response: A related work subsection was added to the introduction and comparative analysis of was added in discussions.

6. Reference section should be strengthened with recent ones. 

Response: Modifications to introduction and discussion sections involved adding more and recent references to the paper. 

Round 2

Reviewer 1 Report

The authors have addressed most of my concerns.

However, I still believe that the paper needs comparison results to prove the validity of the authors` findings. Otherwise, please discuss the reason that no comparison study can be conducted.

Author Response

English language and style are fine/minor spell check required

Response: After modifications, a revision of english was carried out.

The authors have addressed most of my concerns. However, I still believe that the paper needs comparison results to prove the validity of the authors` findings. Otherwise, please discuss the reason that no comparison study can be conducted.

Response: A set of simulations with linear sequence as training structure have been performed on seven ML algorithms: k-nearest neighbours, support vector machine with linear function kernel, support vector machine with radial basis function kernel, random forest, adaptive boosting, naïve Bayes and quadratic discriminant analysis, which were briefly analysed in the discussion section.